# Model-Based Performance Analysis of Membrane Reactor with Ethanol Steam Reforming over a Monolith

**DOI:** 10.3390/membranes12080741

**Published:** 2022-07-28

**Authors:** Ludmilla Bobrova, Nadezhda Vernikovskaya, Nikita Eremeev, Vladislav Sadykov

**Affiliations:** Federal Research Center Boreskov Institute of Catalysis SB RAS, 630090 Novosibirsk, Russia; lbobrova@catalysis.ru (L.B.); vernik@catalysis.ru (N.V.); yeremeev21@gmail.com (N.E.)

**Keywords:** hydrogen separation membranes, catalytic membrane reactors, ethanol stream reforming, mathematical modeling

## Abstract

Membrane reactors (MR) with an appropriate catalyst are considered to be an innovative and intensified technology for converting a fuel into the hydrogen-rich gas with the simultaneous recovery of high-quality hydrogen. Characteristics of an asymmetric membrane disk module consisting of a gas-tight nanocomposite functional coating (Ni + Cu/Nd_5.5_WO_11.25-δ_ mixed proton-electron conducting nanocomposite) deposited on a gas-permeable functionally graded substrate has previously been extensively studied at lab-scale using MRs, containing the catalyst in a packed bed and in the form of a monolith. The catalytic monolith consisted of a FeCrAl substrate with a washcoat and an Ni + Ru/Pr_0.35_Ce_0.35_Zr_0.35_O_2_ active component. It has been shown that the driving potential for hydrogen permeation across the same membrane in a monolithic catalyst –assisted MR is greater compared to the packed bed catalyst. This paper presents results of the study where a one-dimensional isothermal model was used to interrelate catalytic and permeation phenomena in a MR with ethanol steam reforming over the monolith, operating at atmospheric pressure and in the temperature range of 700–900 °C. The developed mathematical reaction–transport model for the constituent layers of the catalyst-asymmetric membrane assembly together with a Sieverts’ equation for the functional dense layer, taking also into account the effect of boundary layers, was implemented in a COMSOL Multiphysics environment. Good agreement with the experimental data of the lab-scale MR with reasonable parameters values is provided. In numerical experiments, concentration profiles along the reactor axis were obtained, showing the effect of the emerging concentration gradient in the boundary layer adjacent to the membrane. Studies have shown that a MR with a catalytic monolith along with appropriate organization of a stagnant feed flow between the monolith and the membrane surface may enhance production and flux of hydrogen, as well as the efficiency characteristics of the reactor compared to a reactor with packed beds.

## 1. Introduction

The use of an expensive hydrogen-fueling network motivated by concerns about environmental issues would lead to high costs in the fuel delivery system. On site hydrogen generation from a hydrocarbon feedstock is considered to be preferable [1,2].

Membrane-based reactors for hydrogen production are among the leading process intensification technologies for small or medium scale applications that open new pathways for both material chemistry and process engineering. The term membrane reactor first began to appear in the literature on chemical technology around 1980 [3]. Although there is no generally accepted definition of a membrane reactor, this term usually refers to membrane devices in which chemical conversion is simultaneously carried out under conditions in which the unique contacting and separating properties of membranes are used. Basically, two configurations of the membrane reactor system are distinguished: in the first case, the reactor and the membrane separation equipment are simply connected in series, while in the second case, the real membrane reactor concept combines the membrane separation process with chemical or biochemical reactions into a single unit. The combination of a membrane separation and a catalytic reaction facilitates process miniaturization, continuous operation and energy saving [4,5].

Membrane reactors with a proper catalyst capable of providing a higher fuel conversion and a membrane possessing permselectivity toward a product gas allow very pure hydrogen to be produced, reducing the down-stream purification load [6,7,8,9,10]. Extensive research has focused on the application of membrane reactors for hydrogen production by the reforming of bio-ethanol, which is an important candidate as a chemical carrier of hydrogen [8,11,12,13,14,15]. The main reaction products are H_2_ and carbon-containing species (CO_2_, CO, CH_4_ and/or C) depending on the catalyst and ethanol-to-water ratio [16,17]. The largest amount of hydrogen is obtained by steam reforming of ethanol, so theoretically 6 moles of hydrogen are formed per mole of ethanol in the feedstock. According to the stoichiometry of the reaction, the molar ratio of water vapor to ethanol (S/E) for the complete conversion of the main reagents into carbon dioxide and hydrogen is 3. According to the thermodynamic study performed by Sun et al. [18], for the reforming reaction with the H_2_, CO, CO_2_ and CH_4_ productions, at about 1000 K more than 5.1 mol of H_2_ (close to 90% of equilibrium yield) can be obtained at S/E > 8. The CO_2_ yield maximizes at about 900 K with S/E > 8. Yet, increasing the S/E molar ratio above 6 has been found to decrease the hydrogen recovery, and more energy is required for feed preparation [15,19].

It has been well studied that the conversion of ethanol and selectivity with respect to various products largely depend on the physico-chemical properties of catalysts, active metal and catalysts supports. Noble metals in catalysts providing a high selectivity to hydrogen are Rh, Ru, Pd and Ir, while among transition metals the best performance ensure Ni, Co and Cu [20]. The order of activity for these metals is Ru > Rh > Ni ~ Ir > Pt > Pd [21]. Despite the high activities and the low tendency of the noble metals to carbon deposition, costs and limited availability of these metals prevent widespread industrial use. Transition metals have shown themselves to be more promising due to their relatively low cost. However, transition metals are more susceptible to coking and deactivate faster than noble metals. On the other hand, the introduction of rare earth oxides into the catalyst support can improve both activity and stability. It is also known that CeO_2_ is effectively used in the development of catalysts for steam reforming of ethanol to produce hydrogen [22,23,24,25,26,27]. Moreover, lanthanum oxide has been reported as an efficient additive to supports providing to catalysts a higher selectivity and stability [28]. Other catalyst types based on doped zirconia [29], alumina [25], zeolites [30], etc. supports [31,32,33] are used in ethanol steam reforming as well.

In the practical design and operating decisions, such multifunctional reactor configuration consists of two reactor volumes, which are the reaction (feed-side) compartment followed by a permeate sweep-side zone. A functionally selective diffusion barrier separating reactor compartments enables hydrogen produced by a catalytic reaction to migrate into the side of a lower hydrogen concentration and be carried out by a sweeping gas, while the retentate stream leaves the feed-side compartment. Mass transport across a membrane module may occur under induced driving forces. The driving force for hydrogen transmembrane permeation is a hydrogen partial pressure gradient at the sides of the membrane. A variety of membrane separation processes are often classified according to their driving forces [34,35,36].

In developing MRs with an integrated catalytic process, the effective characteristics and architecture of the catalyst and the membrane module under identical operating conditions are of utmost importance. Characteristics of a catalyst, such as its design and levels of intrinsic activity and selectivity sufficient to compete with membrane efficiency, are essential for creating a high hydrogen gradient all over the membrane module. The competition between the different phenomena determining the performance of the system, namely mass transport, reaction in the catalytic bed and hydrogen permeation across the membrane may occur. Morever, the catalyst and membrane need to be adapted to provide a low overall resistance to mass transport of hydrogen. Among factors influencing the net driving force of hydrogen transport is the fluid dynamics, which is highly dependent on the chosen experimental reactor configuration and operating conditions [37,38,39].

Clearly, a thorough understanding of all factors that could affect interrelated phenomena of mass transport, catalytic activity and permeation is required when designing a MR. Moreover, these types of reactors may inherently have serious drawbacks such as deactivation of catalyst and membrane due to coke formation, limitations for mass transfer and difficulty in replacing inside materials, etc. [40,41]. A fixed bed of catalyst is the most common approach to the design of reforming reactors for hydrogen production, whereas it suffers from poor heat and mass transfer behavior, high-temperature gradient, catalyst sintering, coke deposition and dust jamming. Catalytic packed beds consisting of particles (typical sizes 1–10 mm) make possible some limitations to hydrogen transport between the bulk of the catalytic bed (where hydrogen -rich gas is produced) and the membrane surface, thus lowering the driving potential for the transmembrane permeation [10,42,43].

Previously, a packed bed membrane reactor for production of hydrogen through ethanol steam reforming has been successfully tested at ambient pressure and the temperature range of 700–900 °C at a laboratory scale [44]. The asymmetric membrane disk module consisted of a gas-tight nanocomposite functional coating (Ni + Cu/Nd_5.5_WO_11.25-δ_ mixed proton-electron conducting nanocomposite) deposited on a gas-permeable functionally graded foam substrates. A packed bed consisting of spherical catalyst particles of 1 mm diameter was placed on the top surface of the membrane module. The best results have been obtained at 900 °C and feed of ethanol/H_2_O mixture in Ar at steam-to-ethanol ratio of four. The overall hydrogen flux was achieved to be about 1.31 Nml cm^−2^ min^−1^.

By using the one-dimensional isothermal reaction–transport model for the constituent layers in the experimental reactor, numerical experiments had been performed to elucidate an impact of the structural parameters of composite membranes on the interrelated catalytic and permeation phenomena in the MR with a packed bed. It was shown that the asymmetric support contributes up to 70% to the overall resistances across the membrane module at the feed gas flow rates of 3 and 10 Nl h^−1^. Transmembrane transport was mainly controlled by chemistry in the catalytic layer located on the surface of the membrane [45].

It is known that structural catalysts (honeycomb monoliths as well as microchannel plates) with a thin catalytically active layer of about 0.01 mm thickness washcoated over metallic substrates provide both a low pressure drop and a low pore diffusion resistance, hence, a high activity. The mini-channels with characteristic diameters between 400 μm and about 1 mm have large surface-to-volume ratios (catalytically active surface area per the catalyst unit volume), which result in superior transfer properties and, consequently, offer a compact and modular solution for the devices. The excellent thermal conductivity of metallic monoliths provides more uniform temperature profiles along the catalysts length/diameters. Advantages such as the ability to intensify catalytic processes by increasing heat and mass transfer or by more precise control of contact times can also improve process efficiency, arising from a specific flow regime that occurs in small channels—nearly plug flow behavior [46,47].

The same membrane module was applied in a lab-scale MR, in which a structural catalytic monolith was installed. In order to evaluate the efficient production of hydrogen and the permeability of the membrane, intensive studies were conducted under various operating conditions concerning temperature, fuel concentration (ethanol and water in Ar) and various molar ratios between ethanol and water [48]. The catalytic monolith consisted of a FeCrAl substrate with corundum protective coating and an Ni + Ru/Pr_0.35_Ce_0.35_Zr_0.35_O_2_ active component. The catalytic monolith was placed above the membrane module and separated from the dense functional layer by a gap junction. Functionally, the gap allowed for the feed stream to flow in a direction perpendicular to the membrane surface, while retentate flow was discharged tangentially along the surface. The experiments were carried out with constant flow rates of 5 Nl h^−1^ for the feed gas (ethanol and water mixture in argon) and of 10 Nl h^−1^ for the Ar sweeping gas. The concentrations of both ethanol and water were systematically varied from 3 to 30 vol. % and from 13 to 80 vol. %, respectively, at the steam-to-ethanol molar ratios S/E = 2, 4, 6. It has been shown that the monolithic catalyst-assisted MR is capable of increasing the driving potential for hydrogen permeation through the same membrane as compared with that of the packed bed catalyst. The maximum flux of 3.03–3.5 Nml H_2_ cm^−2^min^−1^ was obtained at 900 °C, which was nearly independent of the S/E ratio while in the membrane reactor with a catalytic packed bed; this characteristic was calculated to be about 1.31 Nml cm^−2^ min^−1^ [44]. Additionally, an increase in the fuel concentration (ethanol and steam) did not result in improving reactor performance characteristics. Thus, in this mode of operation, hydrogen permeation was limited by passage through the membrane itself. Namely, the diffusion flux through the membrane was no longer governed by the driving partial pressure force, but rather by the ability of the membrane itself to hydrogen mass transfer. The observable phenomenon is a final result of the interplay between catalysis, kinetics, transfer effects and design and operation of the reactor. It makes us curious to understand in detail unobservable critical elements that have important roles to play in the effectiveness of the reaction–separation process in the MR.

The research concept of this paper, based on the intensive experimental study, is to investigate, by conducting a model-based performance analysis, all the main phenomena that determine the performance of the monolithic catalyst-assisted MR, namely design parameters, chemistry, transfer effects and hydrogen permeation across the membrane layers, as well as boundary layer effects. As far as we know, we have not found such a study in the available literature. Simulations and numerical methods are the powerful tools for scientific explanations and predictions, being also an alternative to the techniques of experimental science and observation, especially in cases when phenomena are not observable or when measurements are impractical or too expensive [49].

The computational one-dimensional reaction-transport isothermal model has been implemented in COMSOL Multiphysics. The mathematical model with reasonable parameter values was then verified with experimental data and shown to provide good agreements. Our findings also expand our understanding of the role of gap junction in the reactor, showing it is essential for hydrogen transmitted to the sweep-side compartment. Most significant results are detailed hereafter.

## 2. Reactor Model

Accurate and reliable modeling of a MR calls for a thorough understanding of the processing phenomena that are specific to the particular reactor configuration. Subsequently, in order to reflect the principal features in the mathematical description as accurately as possible the following important information should be provided: the size and general configuration of the reactor and the more important functions and dimensions of its internal structures, operating conditions and the composition of the product emerging therefrom. In this study, we used the following approach: fluxes through all constituent layers were coupled by first defining the concentrations at the interface and the flows on the boundaries inside all assemblies, and then by combining these interface concentrations. Molar flow change due to reaction and membrane transport is accounted for. The developed model is validated by confronting its predictions with data from experimental studies [48].

Moreover, the experimental data have been used to obtain and estimate all necessary parameters to quantify the characteristics for the developed mathematical model.

To capture concentration distributions in axial direction of the experimental MR, a computationally efficient pseudo-homogeneous, one-dimensional isothermal reactor model was developed consisting of the total gas-phase continuity and differential molar balances on species in the constituent layers.

### 2.1. Configuration of Reactor

Detailed information about the design of the experiments, including the experimental setup, the purpose of experimentation, the morphology and the operation of the developed asymmetric supported membrane, consisting of a thin Ni–Cu alloy–Nd tungstate nanocomposite dense permselective layer deposited on a hierarchically structured asymmetric support, has been reported in previous papers [44,45]. The main assumptions performed in the simulations of the MR with the monolith catalyst are summarized below: design, kinetics, and transfer effects. 

Detailed information about the experimental setup and disk-shaped reactor with axis-symmetric flows with ethanol steam reforming over a metal honeycomb catalyst has been reported in detail in the previous paper [48]. The catalytic monolith is also, as in the case of the packed bed membrane reactor [45], located above the membrane module. A sketch of the MR is shown in Figure 1, showing all essential constructive details of the experimental reactor that is under present numerical study.

The use of a well-insulated cylindrical furnace ensures constant temperature in the reactor system. In this reactor configuration, the monolith and membrane surface are linked via a gap junction. It is the main difference between the monolith bed and the packed bed membrane reactors. The constructive gap, which separates catalyst and membrane, allows for the hydrogen-rich gas from the catalyst to flow in a direction perpendicular to the membrane surface, while retentate flow is discharged tangentially along the surface out of the reactor system. Hence, the experimental reactor utilizes a stagnation flow configuration in the constructive axial distance between the honeycomb catalyst and frontal surface of the membrane.

The operation at the water/ethanol molar ratio of 4 in the fuel mixture was found to be beneficial for ethanol conversion and hydrogen separation within an operating temperature range of 700–900 °C. These operation parameters were taken as a reference case in the simulations.

To verify a developed model and simulate the reactor behavior for different operational cases, it is necessary to have a reference case. The general parameters for the reactor and experimental conditions (reference case) are given in Table 1.

#### 2.1.1. Catalyst Bed Configuration

The performance of the catalytic monolith is to a large extent determined by its structure, morphology and porosity. A metal support is formed by stacking flat and corrugated foil (Kanthal FeCrAl alloy) bands (each approximately 120 µm in thickness) and winding them into an Arkhimedes spiral (Figure 2). The corrugated foil had a wave height of 0.7 mm at a pitch of 2.5 mm, so the entire cylindrical honeycomb substrate with outer diameter of 24 mm had 12 layers of flat and corrugated foils in the cross section. The support was coated by γ-Al_2_O_3_ (10 wt. %) followed by the catalytically active composition, 10 wt. % Pr_0.35_Ce_0.35_Zr_0.35_O_2_ + 5 wt. % Ni + 1 wt. % Ru as described elsewhere [48]. The geometrical properties of the catalytic monolith together with the feed gas composition for the reference case are listed in Table 1.

#### 2.1.2. Design of Membrane Module

The morphology and geometry of membrane module were discussed in detail in the previous paper [45]. The asymmetric supported membrane consists of a gas-tight Ni–Cu alloy–Nd tungstate nanocomposite dense functional layer deposited on a gas-permeable Ni-Al hierarchically structured asymmetric substrate (Figure 3) [44]. The Nd_5.5_WO_11.25-δ_ (NW) powder synthesized by the mechanical activation as described elsewhere [50] was characterized as a single-phase defect fluorite. The Ni–Cu alloy was synthesized by modified Pechini route using a fluidized bed reactor for subsequent thermal treatment of products of polymeric precursors decomposition in Ar + H_2_ streams. The obtained nanoparticles were put into isopropanol to avoid oxidation. The Ni–Cu (30 wt. %)/Nd_5.5_WO_11.25-δ_ nanocomposite was prepared by ultrasonic dispersion of Ni–Cu alloy and NW powder suspension in isopropanol with the addition of polyvinyl butyral. The membrane parameters are given in the Table 2.

Powder Metallurgy Institute (Minsk, Belarus) has provided functionally graded substrates. The method for making gas-permeable structures involves applying two thin low-porosity layers onto a thick Ni/Al foam. The tough foam is made from alumina-silica ceramics. In this process, 30 PPI polymeric foam is used as a template. Macroporous ceramics is obtained by 3-times impregnation of foam polyurethane with alumosilicate suspension followed by centrifugation, drying and sintering at 1350 °C. In addition, Ni covering of alumina-silica ceramics is done by electrolysis followed by drying and sintering at 1000 °C in cracked ammonia. Ni:Al_2_O_3_ weight ratio is obtained to be 2-2.2:1. Formation of two thin layers onto the Ni/Al substrate have been done by the use of dual doctor blades in series. The compaction of layers has been performed via sintering of the specimen at 1000 °C in cracked ammonia followed by in-pack aluminizing as described in Ref. [51]. The resulting gas-permeable substrate has a composite structure, in which the upper porous layer has pores of a smaller diameter than the intermediate porous layer on the foam substrate. In terms of pore size distributions and porosity, the three-layer structure of the gas-pearmable support is clearly quantified and designated in Table 1. Quantification of the constituent particles and pore systems has been done via the image analysis with a Joyce–Loebl Mini Magiscan (Joyce-Loebl, Ltd., Gateshead, UK) computerized image analysis system [52,53], and with a vector program, by applying an appropriate spatial calibration in a series of parallel and perpendicular lines along the image axes in the reference area.

#### 2.1.3. Gap Flow Characterization

The reactor configuration introduces a space between the catalytic monolith and the membrane dense surface, in which case the hydrogen-rich product stream from the catalyst layer must pass across the gap before it reaches on the membrane surface. That is, the dense surface of the membrane is perpendicular to the flow, thereby creating a stagnation region, in which the feed flow velocity vanishes, and the gas continues flowing out radially outward to the exhaust through an annular space around, causing the formation of a boundary layer adjacent to the stagnate surface. By convention, the value of the boundary layer thickness is defined as a height needed to obtain 99% of the mainstream velocity.

It is known that the stagnation region encounters the highest pressure, the highest transfer rate and the highest rates of mass exchange and deposition to a flat surface [54,55]. Under uniform distribution and effective and stable contact with the membrane surface, hydrogen from the product gas selectively permeates into the sweep-side compartment. The axisymmetric stagnation flow geometry is a desirable flow scenario for controlled measurements [56,57], as it ensures that an active surface sees at least the same gas-phase composition. The constructive and the operating parameters in the stagnant flow arrangement, affecting the experimental information, were investigated quantitatively by F. Zanier et al. [58]. It was shown that the flow structure may even become the controlling factor determining the global activity of disk-shaped, flat active surfaces.

Since the gap is full of stagnant hydrogen-rich gas, there should be a reduction in the mass transfer resistance and, thus, a higher hydrogen flux compared to membrane reactors with packed beds. Understanding whether our experimental evidence supports the conclusion is a goal of the present paper.

In the modeling approach, a steady similarity numerical solution has been applied to describe the axisymmetric stagnation fluid flow toward the membrane.

### 2.2. Model Description

The mathematical model of the membrane reactor utilizes a stagnation flow configuration in the gap between the catalytic monolith and dense surface of the membrane module. The molar balances comprise axial mass transfer by convection and axial dispersion as well as mass transfer through the phase interfacial surface area.

The following assumptions have been made in the formulation of the mathematical model:Both the reaction and transport are conducted isothermally. This assumption for the endothermic reaction of ethanol reforming signifies that sufficient heat is supplied to maintain an approximately constant reaction temperature.The internal mass transfer resistance of the catalytic washcoat over the metallic monolith is neglected.Negligible pressure drop occurs across the membrane reactor.The permeation of hydrogen through the dense layer of the membrane module follows Sievert’s law.Molecular and Knudsen diffusions control in the hydrogen and argon transport through the gas-permeable layers of the membrane support.A perfect-gas equation of state relates the density, pressure, temperature and composition.

Important constitutive equations for the reaction kinetics, membrane flux and dispersion coefficients are discussed in following sections. A set of equations for the feed-side is coupled through the hydrogen permeation flux depending on the partial pressures on both sides of the dense permselective layer of the membrane module. As there are not significant pressure gradients, the gas flow has been modeled considering a constant gas density in each layer of the membrane reactor (Table 1).

#### 2.2.1. Feed-Side Compartment Model

The model equations and the requisite boundary conditions have been listed in Table 3 and Table 4.

It is believed [59,60] that in reactors with the separation option, reduced Reynolds numbers (*Re*) are desired, indicating highly ordered laminar flow and a low Péclet number (*Pe* of less than or around 1), when diffusive transports between phases are preponderant. Such is the case with the experimental MR operating conditions for the reference case (Table 4). Thus, at *T* = 900 °C, the Reynolds number is about 1800, while the Péclet number is 0.66, assuring laminar flow conditions. Chemical reactions result in production or consumption of species (*R_i_*), which is modeled as a molar source or sink for the *i*-th specie. A continuity equation for species molar balance in the feed-side compartment contains partial derivatives of molar flow rates and species concentrations with respect to the axial position.

##### Reaction Description

The steam reforming of ethanol is assumed to include a stepwise reaction scheme, see Table 4 [61]. The observed reaction rates depend on the concentration of the reactants, the temperature and the catalyst external surface area. Effective parameters and variables have been evaluated in an earlier experimental study of ethanol steam reforming performed over the 5 wt.% Ni + 1 wt. % Ru/Sm_0.15_Pr_0.15_Ce_0.35_Zr_0.3_O_2-__δ_ catalyst [44].

##### Dispersion of Mass

It is well known [59] that the complexity of the molecular transport processes does not allow a purely theoretical fundamental approach in analysis of the diffusive transport. In the numerical simulations, transport coefficients (the diffusion coefficients, viscosity coefficients etc.) are calculated from the transport coefficients for the individual species (Table 5). For laminar flow in tubular pipes (*Re* < 2100), Taylor [62] and Aris [63] established a relation between the molecular diffusion coefficient, the hydraulic diameter and the average velocity in the channel. For the contribution of molecular diffusion, an effective diffusion coefficient was calculated with the Wilke equation [64], where binary diffusion coefficients *D_ij_* ([m^2^/s]) (Table 5) were calculated by using Fuller-the Schettler-Giddings empirical Eq. [65]. There is considerable uncertainty about hydrodynamics in the feed-side and effective diffusivities of species, which warrants the use of fitting parameters for dispersion coefficients, when agreement with experimental data is desired. 

##### Stagnate Flow in the Gap

The monolithic catalytic structure imposes a well-defined plug-flow condition for the product gas at the outlet ${F_{f}|}_{z=h_{mth}}^{+}$, which stagnates on the surface of the dense layer. The thickness $\delta_{h,m}$ of the laminar boundary layer developed over a flat dense surface is the result of the interplay between convective and diffuse fluxes in the gap. That is, diffusive effects only become important in a thin region near the membrane surface. The change of hydrogen concentrations on each side of the stagnant boundary layer is influenced not only by driving force, but also transport properties in the membrane itself. Hence, the feed composition is changed in this intermediate region resulting from the diffusion of hydrogen to the membrane surface across the boundary layer, and the concentrations decrease from their feed stream values to that of the retentate composition-controlled values. 

The retentate product $F_{r}$ depends on the net production rates of the species produced ${\left(F_{f}\right)}_{z=h_{mth}}$ over the monolithic catalyst and hydrogen flux permeated $\left(F_{H_{2},perm}\right)$ through the membrane (Table 3). When a concentration gradient exists, the species tend to flow in a direction such as to reduce the concentration gradient. The mass transfer coefficient *β*, also known as the transfer velocity, is used to characterize transfer of a substance through another on a molecular scale due to concentration difference or gradient. Mass transfer rate is dependent on the thickness of the boundary layer.

The interfacial effect on diffusion resistance in the stagnation region due to changing geometrical and operational parameters should be introduced obviously. Hiemenz [66] first studied the steady flow in the neighborhood of a semi-infinite wall. For the axisymmetric stagnation fluid flow, a steady similarity numerical solution has been applied quite often [54,55,67]. A measure of a change of flow conditions near the stagnation surface is determined as a stagnation velocity gradient $u\left[\frac{1}{s}\right]$ [68,69]. Expression for the boundary layer thickness in a stagnation-flow given in Table 5 as the square root of the reciprocal value of the velocity gradient [70] takes into account a change velocity with the respect of the gap geometry and gas flow rate.

The mechanism by which hydrogen diffuses through the boundary layer in the gas mixture due to the concentration gradient is exactly analogous to that by which heat diffuses. By adopting the approach of non-dimensional analysis, the mass transfer coefficient *β* is defined in a manner analogous to the heat transfer coefficient in terms of the Reynolds and Schmidt numbers. A theoretical correlation for predicting the heat transfer near the forward stagnation point assuming laminar, incompressible, low-speed flow is used [68]. When used in connection with mass transfer, the Prandtl number is replaced by the Schmidt number, *Sc*, which expresses the ratio of the momentum diffusivity to the mass diffusivity. The stagnation Reynolds number, *Re*, was based on the diameter of the impingement membrane surface, *d_m_*, and the stagnation velocity gradient.

#### 2.2.2. Sweep-Side Reactor Model

##### Dense Layer

Mass transport through dense membranes is a purely diffusive process [71]. Permselectivity is the intrinsic property of the active material, while the driving force for hydrogen transport across the dense membrane is caused by the difference of hydrogen partial pressure on each side. Sieverts’ Law, which identifies the difference of the square roots of the partial pressure of the hydrogen as the the driving force of the permeation, is a temperature-activated phenomena. It is valid with an underlying assumption that the surface coverage is low when interfacial equilibrium is achieved and the rate-limiting step is an atom diffusion through the membrane. These criteria are satisfied in most cases when the temperature is relatively high [72]. Obviously, the steady state of the hydrogen flux through the dense layer of a thickness *h_dm_* being described by Sieverts’ law is equal to the permeate flux through the adjacent boundary layer (parity–flux equation in Table 6). The permeation rate is limited by the concentration of hydrogen on the gas phase /dense layer interphase $x_{H_{2},dm}$. An explicit solution for $\sqrt{x_{H_{2},dm}}$ is attained from the parity–flux equation (Table 6).

It is known that values for the hydrogen flux and the hydrogen permeation coefficient can be estimated more precisely only for given conditions. Therefore, for predicting the hydrogen flux, the apparent gas permeability is useful to quantify for certain operating regimes [73,74]. Corresponding parameters in the expression for the hydrogen permeation flow rate $\left(F_{H_{2},perm}\right)$, the apparent activation energy, and the permeability were obtained from the temperature-dependent conductivity study [75] and the fitting of this equation to the reference case experimental data [48].

##### Gas-Permeable Layers

The mathematical model for the porosity graded substrate was described in detail in a previous article published by the author [45]. The gaseous mixture inside the gas-permeable layers is a binary mixture of argon and hydrogen. The hydrogen gas transport inside is governed by Fick’s diffusion with effective binary diffusion coefficients. The linkage between the layers requires the equality of the fluxes at each boundary (Table 6). The effective pore diffusivity in the foam layer is taken as a function of the structural parameters of the heterogeneous porous medium. Hydrogen diffusion resistance accounted for with the specific mass transfer correlation for the foam configurations [76,77,78].

##### Sweep Compartment

Using the ideal gas law and assuming a perfectly mixed gas, the molar flow rates for the species H_2_ and Ar in the sweep gas compartment are related to the volumetric gas flow accounting for a change in the total molar flow rate of the sweep gas at a given temperature (Table 6). The unknown variable $x_{H_{2},sw}$, the molar fraction of the hydrogen in the permeate flow leaving the sweep compartment, is yielded at steady state to be the real solution of a quadratic.

### 2.3. Numerical Solution Strategy

The scaled reactor model equations for the constituent layers of the experimental membrane reactor equipped with the catalytic monolith (Table 1) with the boundary conditions have been implemented in COMSOL Multiphysics 4.3a software. The numerical model has been executed and solved simultaneously in the domains along the full length. The finite-element method for numerical solutions of differential equations being used employs a uniform (fine) mesh and error control in the domains. A numerical problem arises from the initial values in the feed, which generates a division by zero in the reaction rate equations. This problem has been sorted out by using very small values for the mole fractions of the generated species at the inlet. The solver, an implicit time-stepping scheme, is well suited for solving stiff and nonstiff nonlinear boundary value problems. At the end of the solving process, species concentrations and fluxes are known at each axial location within every reactor’s domains. The physical properties are determined by the local composition and temperature, which are affected by the chemical reactions and the permeation flux. A fast and unconditionally stable solution is provided, when solving the species molar balance. 

## 3. Results and Discussion

### 3.1. Base, Reference Case

#### 3.1.1. Verification of the Model Formulation and Computational Simulations

The feasibility of performing ethanol steam reforming over a monolith in the lab-scale MR in isothermal mode has been investigated extensively at different operating conditions regarding the temperature, the fuel concentration and the different molar ratios between ethanol and water [48]. An isothermal condition in a reactor is the most ideal mode of operation, because the constant temperature along both a catalyst and a membrane is advantageous for the life duration and stability. The reactor operation in the temperature range of 700–900 °C with the water/ethanol molar ratio of 4 in the feed was found to be quite sufficient in terms of the ethanol transformation into syngas and CO_2_ as well as the amount of hydrogen produced from 1 mol of ethanol. These operation parameters were taken as a reference case in the simulations.

The most important question in any simulation is the reliability and validity of an assessment tool and whether we can get more out of our model by tuning the model parameters. The applicability of the one-dimensional model formulation and computational simulations are verified and validated by direct comparison of model results with existing measurements of interest. In Figure 4, the results of the simulation for the hydrogen flux and concentrations of species are compared to the known values in the experimental study of the reference case. The operating conditions and general parameters for the reactor internals for the base (reference case) have been listed in Table 1.

Under the examined operating condition, correction factors were applied to the theoretically calculated beta values (feed-side mass transfer coefficients in Figure 5) in order to match satisfactorily the experimental data in the studied temperature range with the same magnitude of permeability ($\Theta =1.4801\times 10^{-2}$ mol m^−1^s^−1^atm^−0.5^) used in modeling.

Consistent with the previous results of modeling the MR with a packed bed catalyst, our simulations are in reasonably good agreement with the experimentally deduced species concentration in both the feed- and sweep-side and the hydrogen permeation flux. In particular, by considering the effect of the temperature shown in Figure 4, the average error between the experimental and the modeling data for the hydrogen fluxes is about 5%, while a highest deviation of about 8% is observed for hydrogen concentration in the retentate gas at 700 °C. 

The concentration distribution along the reactor internals affects the overall hydrogen permeation flux, which, in turn, depends on mass transport parameters: reaction kinetics, Péclet number and effective diffusivity in the constituent layers [79,80]. Understanding all these phenomena is important when studying factors that direct the hydrogen removal.

#### 3.1.2. Performance Analysis

Generally, comparative analysis of MRs is performed by using characteristics of recovery, yield of hydrogen, fuel conversion and resistances to internal mass transfer of a permeating component. The details of introducing a structural catalyst in the MR reactor instead of a packed bed on the overall efficiency of the integrated reaction–separation process were discussed in detail in the previous paper [48]. It was shown that the reactor operating with the catalytic monolith showed better performance in terms of both hydrogen recovery and the yield with respect to the packed bed catalyst. Thus, a comparative study showed that in the case of a packed bed, an increase in temperature from 700 °C to 900 °C leads to an increase in the yield of hydrogen by almost 40% (from 24% to 33%), while the yield for the monolith practically does not change and averaged 37%. Hydrogen recovery for the monolith was about 10% higher in all operating temperatures.

The resistance concept proposed by J.M. Henis and M.K. Tripodi [81] gives insight and understanding about which of the reactor internals is controlling the total flux. Irrespective of the transport mechanisms, a total transport resistance consists of the layers resistances in series and includes also resistances by the gas film on both sides of the membrane assembly [45,82,83,84]. A layer with the highest concentration gradient would provide the major mass transfer resistance to the overall permeation process.

Typical concentration profiles, as those calculated for the reference case at T = 800 °C, Figure 6a, evolve throughout the reactor length at isothermal operation. First, the hydrogen concentration increases up to a maximum value (29.8 vol. %) due to the high ethanol reforming reaction rate and then, after its maximum value, the concentration gradually decreases down towards the membrane surface up to 23 vol. %. The concentration gradient of about 3.6 vol. % is established across the boundary layer adjacent to the membrane. The percentage values of hydrogen recovery and yield estimated by experiments at 800 °C give 47% and 44%, respectively.

The hydrogen removal results in an increase in the overall mole fraction of other species and a depletion of hydrogen in the retentate gas. This causes a concentration gradient build up in the boundary layer. The polarization reduces the overall efficiency of a membrane separation [79]. The polarization resistance of the membrane assembly at different temperatures in the reference case are shown in Figure 6b. The slope of the curves is mostly dependent on the temperature and on the permeability of the membrane layers. High temperature is beneficial to syngas formation and membrane permeability. When the operating temperature increases from 700 up to 900 °C, the hydrogen flux increases by ~35% while the overall resistance to the hydrogen mass transfer decreases by ~38% as the temperature increases. The values with high impact on the overall efficiency are at least in the same order of magnitude. It means that resistance values have to be estimated in a simulation procedure. The derived model allows describing resistances to the hydrogen mass transfer as a function of the operating conditions and the structural properties of each single layer.

The MR components that contribute to the hydrogen permeation resistance performance are boundary layers from both sides of the membrane module and its constituents —dense, powder, intermediate and foam layers. The contribution to the overall resistance of the components is illustrated in Figure 7.

It can be seen that the hierarchically structured asymmetric support, which is required for desirable industrial applications of membranes, demonstrates a similar extent of resistance to hydrogen transport in the constituent gas-permeable layers for both the configuration of the catalyst and for the different feed-flow rates, while the variable contribution of the permselective layer is observed. In the stagnant flow condition, molecular hydrogen is able to pass easily to the dense layer of the membrane. However, the transmembrane potential, which can be characterized by the concentration gradient across the membrane and by the intrinsic permeability, fails to permeate more flux. Indeed, depending on the conditions at the entry (feed side) and exit (sweep side) faces of the membrane, a concentration gradient establishing through the membrane module gives rise to the membrane permeation. Yet, the maximum flux was detected at 900 °C to be about 3.03–3.5 Nml H_2_ cm^−2^ min^−1^ [48]. Thereby, the stagnation flow mode is supposed to elicit a limit of the membrane permeation, when the diffusion flux through the membrane is no longer governed by the driving partial pressure force, but rather by the intrinsic features of its constituents. Thus, the permeation rates are higher for the monolith-assisted MR than for the cases of the packed bed and limited by the permeation ability of the dense membrane layer.

### 3.2. Parametric Study

As it is drawn in Figure 6a, the nonvanishing gradients of concentrations in the gap are restricted to the boundary layer. This is due to the convective nature of the mass transport in the gap. The gas streams toward the dense layer and the permeation process at the surface have no influence on the fields far away. Species move with the flow, which dominates over any non-convective transport. In this flow configuration, only when the axial velocity approaches zero near the surface, the diffusion becomes of similar size as the convective transport [58]. The remaining retentate products are transported away sufficiently quickly.

We modeled various distances between the monolithic catalyst and the membrane surface, as well as feed rates, to determine their effect on permeability characteristics. As expected, the thickness of the boundary layer affects the hydrogen diffusion rate through it. Figure 8a, left panel, shows that by enlarging the gap above the membrane surface from 4 to 15 mm, the thickness of the boundary layer increases from 1.28 mm to 2.48 mm at 700 °C, and from 1.46 mm to 2.83 mm at 900 °C. This leads to a decrease in the mass transfer velocity of hydrogen β (at 700 °C from 0.35 to 0.09 m/s, at 900 °C from 0.81 to 0.21 m/s) with a corresponding decrease in the hydrogen flux (about 3–4%). Indeed, a smaller distance between the monolith and the membrane can lead to compressing the boundary layer and, thereby, can provide better diffusion of hydrogen into the dense layer. 

Higher feed flow rates have the same effect on the boundary layer: the thickness of the boundary layer on feed-side decreases. However, this leads to lower values of both the hydrogen flux and hydrogen concentration in the sweep compartment, as shown in Figure 8b, due to an increase in the hydrogen concentration in the retentate flow coming out of the gap space. The higher the feed rate, the higher the negative effect of a larger gap height above the membrane surface can be.

The higher hydrogen flux in the case of the MR with a monolith may also be due to the influence of the boundary layer phenomenon after the impingent of the feed stream (product gas from catalytic monolith), which affects the corresponding driving forces of interfacial hydrogen transfer rate in the membrane. The effect of the operating temperatures on the driving force, expressed here as the difference between the concentrations on each side of a boundary layer adjacent to the membrane surface and the magnitude of the corresponding permeation fluxes, is shown in Figure 9. It can be seen that in the case of the stagnate boundary layer, the driving force is about an order of magnitude higher, and accordingly the permeation flux is 30–40% greater. It is obvious that the hydrogen flux through the membrane is dictated by the number of molecules impinging and interacting with the interfacial surface under the boundary condition being formed. 

At the same time, the stagnate boundary layer phenomenon must be properly taken into account for the specific reactor configuration and operating conditions; otherwise incorrect conclusions about the technological parameters will be drawn. For example, if the feed flow is moving at a very high speed, it may have some kinetic energy, and this may lead to undesirable effects on the penetration rate.

## 4. Conclusions

In this study, the conceptual feasibility of a membrane reactor equipped with a catalytic monolith for ethanol steam reforming has been investigated by a detailed numerical simulation using a 1-D reactor model. The formulated mathematical description takes into account the design parameters of the MR, the chemistry and transfer effects in the monolith, the hydrogen permeation across the membrane constituent layers together with a Sieverts’ equation for the functional dense layer as well as the boundary layer effects. The scaled reactor model equations were implemented in a COMSOL Multiphysics environment. Good agreement with the experimental data of the lab-scale MR with reasonable parameters values is provided. In the numerical experiments for the MR operating at atmospheric pressure and in the temperature range of 700–900 °C, the concentration profiles along the reactor axis were obtained, showing the effect of the emerging concentration gradient in the boundary layer adjacent to the membrane.

A parametric study was carried out to determine the influence of the distance between the catalyst monolith and the membrane surface, as well as the feed rate, in order to determine their influence on the permeability characteristics. Polarization resistance, which reduces the overall membrane separation efficiency, has also been elucidated for the reference case and compared with the packed bed MRs. 

The main findings are as follows:the use of a catalytic monolith with stagnant flow between the catalyst and membrane surface may increase both the production and flux of hydrogen, as well as reactor efficiency characteristics compared to a packed bed MR;the stagnate feed flow configuration must be adequately taken into account for the specific reactor design and the operating conditions; otherwise incorrect conclusions about the technological parameters will be derived. Thus, increasing the distance between the monolith and the membrane surface from 4 to 15 mm increases the thickness of the boundary layer from 1.28 mm to 2.48 mm at 700 °C, and from 1.46 mm to 2.83 mm at 900 °C with a corresponding reduction in the hydrogen flux (about 3–4%);at higher feed flow rates, the thickness of the boundary layer on the feed-side of the membrane decreases. However, in this case, lower values of both the hydrogen flux and hydrogen concentration in the sweep compartment are obtained, due to an increase in the hydrogen concentration in the retentate flow leaving the gap space. The higher the feed rate, the higher the negative effect of a greater gap height above the membrane surface can be;the hierarchically structured asymmetric membrane support, which is often necessary for industrial applications, demonstrates a similar extent of resistance to hydrogen transport in the constituent gas-permeable layers of the membrane used in the study of MRs with monolith and packed bed catalysts and for different feed-flow rates. A variable contribution of the permselective layer is observed, and the permeation rate in the MR with the monolith is limited by the permeability of the dense membrane layer.

This study proved that catalytic monoliths could be effectively used inside MRs to produce hydrogen. Comparative studies of hydrogen mass transfer through the same membrane module using an in series model resistance showed a stronger effect for the monolith-assisted MR with stagnant flow over the membrane when compared to a packed bed. In this case, all mass transfer resistance will lie in the membrane module itself, especially in the permselective layer, and almost not be in the boundary layers.

## Figures and Tables

**Figure 1 membranes-12-00741-f001:**
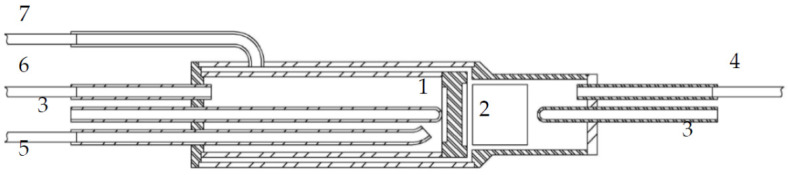
Sketch of a membrane reactor with catalytic monolith: 1—asymmetric supported hydrogen separation membrane, 2—catalytic monolith, 3—thermocouples pockets, 4—feeding tube, 5—Ar sweep gas tube, 6—permeate outlet tube, 7—retentate gas tube.

**Figure 2 membranes-12-00741-f002:**
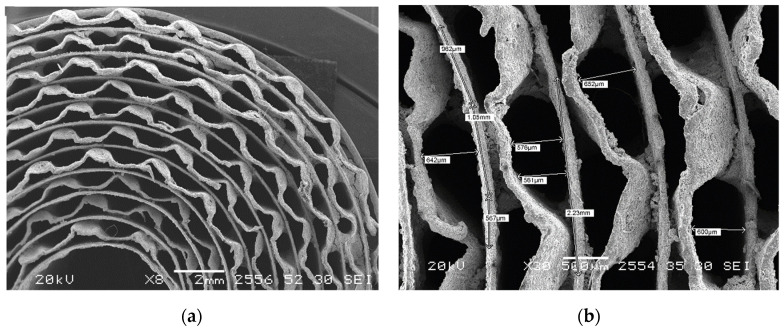
Pictures of the experimental catalytic monolith: (**a**) a general view, (**b**) a fragment with dimensions.

**Figure 3 membranes-12-00741-f003:**
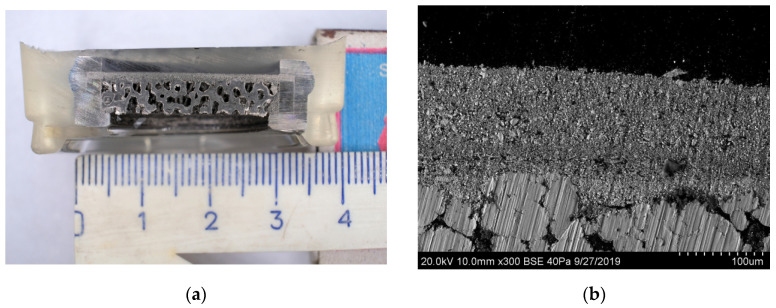
Cross-sectional images of the layer-by-layer assembled membrane module (**a**) and SEM microgragh of the inlet area (**b**).

**Figure 4 membranes-12-00741-f004:**
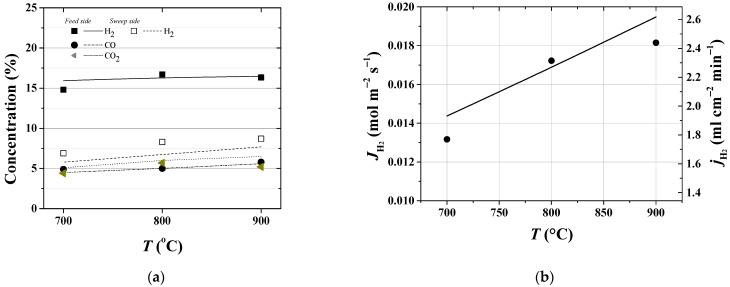
Effect of the operating temperature (**a**) on species concentration (dry basis) and (**b**) on hydrogen permeation flux at the retentate and sweep streams. Data points correspond to the measurements of gas phase average composition.

**Figure 5 membranes-12-00741-f005:**
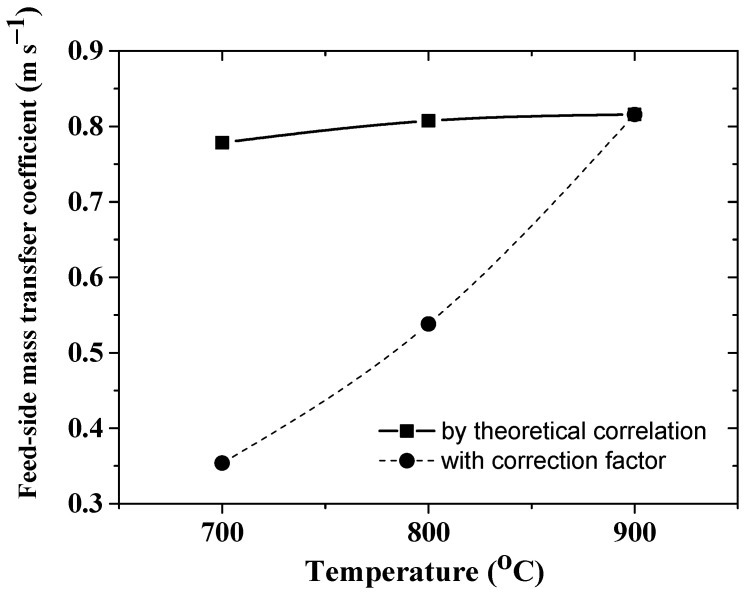
Temperature curves of feed-side mass transfer coefficients: a theoretically calculated one and those with application of correction factors.

**Figure 6 membranes-12-00741-f006:**
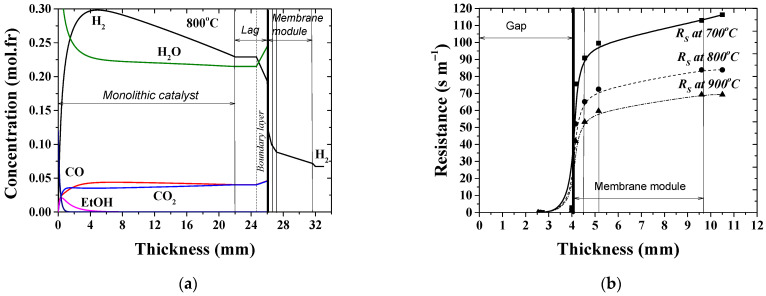
Concentration (**a**) and hydrogen mass transfer resistance (**b**) profiles along the monolith-gap-membrane assembly in the MR at *Pe* = 0.76; *T* = 800 °C; Table 1.

**Figure 7 membranes-12-00741-f007:**
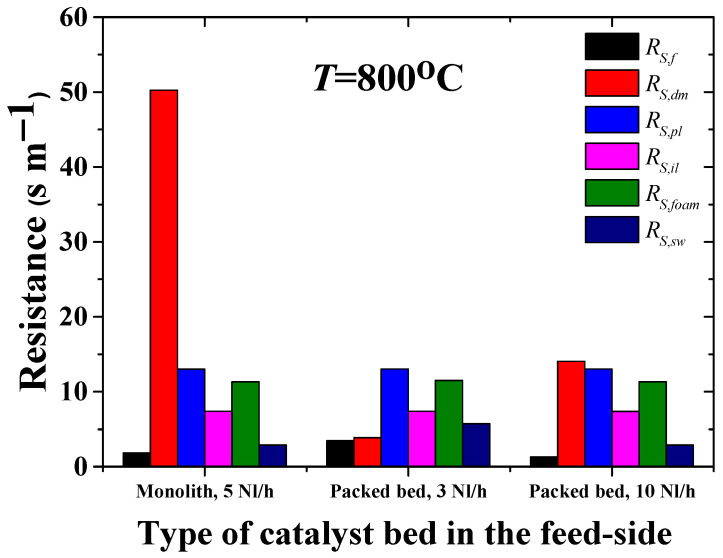
Comparison of the MRs with packed bed and monolith at different feed flow rates in contribution of each individual layers to the overall hydrogen mass transfer resistance.

**Figure 8 membranes-12-00741-f008:**
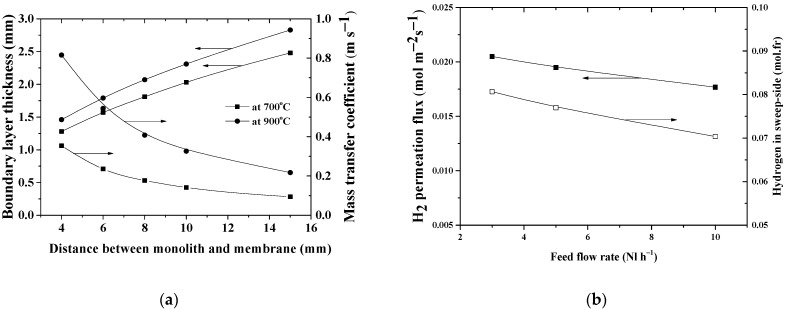
Effect of the distance above the membrane surface (**a**) and the feed flow rate (**b**) on the permeation phenomena.

**Figure 9 membranes-12-00741-f009:**
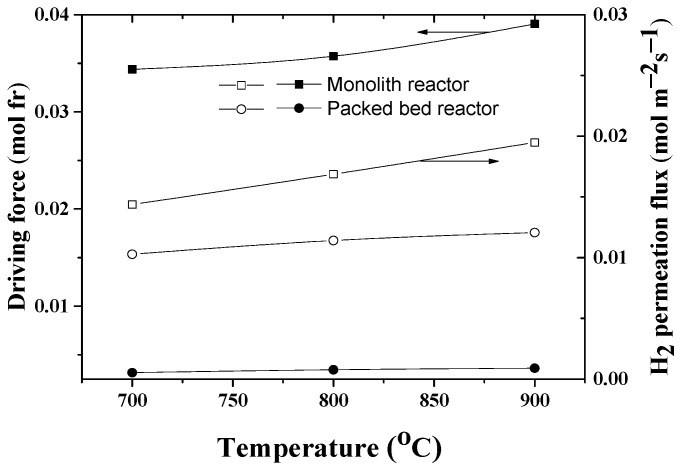
Effect of the temperature on the driving force (difference between concentrations on each side of a boundary layer adjacent to the membrane surface) and hydrogen permeation flux in the MRs with monolith (Feed: 5 Nl h^−1^) or the one with the packed bed (Feed: 3 Nl h^−1^).

**Table 1 membranes-12-00741-t001:** Structural parameters of the constituent layers of the experimental membrane reactor assisted with the catalytic monolith and experimental conditions (reference case) used in simulations.

**Variable**	**Units**	**Value**
**Feed-side**		
**Catalytic monolith:**		
Height (*h_mth_*)	mm	22
Diameter (*d_mth_*)	mm	24
Cross section area (*A_mth_*)	mm^2^	452.16
Equivalent channel diameter (*d_h,mth_*)	mm	0.6926
Porosity (*ε_mth_*)	(-)	0.58
Volumetric surface area (*S_V,mth_*)	m^2^ m^−3^	3355
Flow rate of ethanol	Nl h^−1^	0.6
Flow rate of steam	Nl h^−1^	2.4
Flow rate of argon	Nl h^−1^	2.0
**Gap with stagnate flow:**		
height (*h_gap_*)	mm	4
**Sweep-side**		
**Membrane module:**		
Dense layer		
Thickness (*h_dm_*)	mm	0.15
Powder layer		
Thickness (*h_pl_*)	mm	0.4
Particle size (*d_pl_*)	mm	0.072
Hydraulic pore diameter (*d_pore,pl_*)	mm	0.012
Porosity (*ε_pl_*)	(-)	0.2
Tortuosity(*τ_pl_*)	(-)	4.2
Volumetric surface area (*S_V,pl_*)	m^2^ m^−3^	66,667
Intermediate layer		
Thickness (*h_il_*)	mm	0.6
Particle size (*d_il_*)	mm	0.061
Hydraulic pore diameter(*d_pore,il_*)	mm	0.027
Porosity (*ε_il_*)	(-)	0.4
Tortuosity (*τ_il_*)	(-)	3.4
Volumetric surface area (*S_V,il_*)	m^2^ m^−3^	59,259
Foam layer		
Thickness (*h_foam_*)	mm	4.5
Cell diameter (*d_cell_*)	mm	2.2
Hydraulic pore diameter (*d_p,foam_*)	mm	1.006
Porosity (*ε_foam_*)	(-)	0.75
Tortuosity (*τ_foam_*)	(-)	1.42
Volumetric surface area (*S_V,foam_*)	m^2^ m^−3^	1395.4
Flow rate of argon	Nl h^−1^	10

**Table 2 membranes-12-00741-t002:** Morphological and structural characteristics of the prepared asymmetric membrane module. Reprinted from Ref. [45] under the CC BY 4.0 License.

Layer	Composition	Thickness (µm)	True Density (g cm^−3^)	Particle Size ^b^ (µm)	Pore Diameter ^b^ (µm)	Porosity ^c^ (%)
*Dense layer*	Ni–Cu/Nd_5.5_WO_11.25-δ_	93.3–115 (center); 194–256 (edge)	6.6	0.045 for Ni–Cu, 0.1–1 for Nd_5.5_WO_11.25-δ_	15 (*x*) 42 (*y*)	~4
*Powder layer*	Ni-Al	380–440	~7	65 (*x*) 81 (*y*)	12 (*x*) 11 (*y*)	12–14
*Intermediate layer*	Ni-Al	400–1300	5.34	45 (*x*) 50 (*y*)	27 (*x*) 27 (*y*)	27–32
*Foam layer*	Al_2_O_3_-SiO_2_ foam with Ni-Al coating	4500–5000	4.63	2400(*x*) ^a^ 1800 (*y*) ^a^	1000 (*x*) 1100 (*y*)	38–40 83 ^d^

^a^ Cell diameter; ^b^ *x* axis is parallel and *y* axis is perpendicular to the membrane surface; ^c^ Quantification with vector program and by image analysis; ^d^ Overall porosity including pores in Al_2_O_3._

**Table 3 membranes-12-00741-t003:** Governing equations for the feed-side reactor.

Monolith bed	
Component molar balance $\varepsilon_{mth}\rho_{tot}\frac{\partial x_{i,f}}{\partial t}+\frac{1}{A_{mth}}\frac{\partial (F_{f}\cdot x_{i,f})}{\partial z}-D_{i,mth}\rho_{tot}\frac{\partial^{2}x_{i,f}}{\partial z^{2}}=S_{V,mth}R_{i}$, where $R_{i}=\sum_{j=1}^{4}\nu_{ij}r_{j}$	
Conservation equation for the change in the total molar flow rate $\frac{dF_{f}}{dz}=\sum_{i}\frac{F_{f,i}}{dz}=2A_{mth}S_{V,f}\cdot \left(r_{1}+r_{2}+r_{4}\right).$	
Boundary conditions	
Inlet (*z* = 0):	${x_{i,f}|}_{z=0}=x_{i,f}^{0}$ $F_{f}=F_{f}^{0}$
Outlet (*z* = *h_mth_*): $i\neq H_{2}$	${\left(F_{f}\cdot x_{i,f}-A_{mth}D_{i,mth}\rho_{tot}\frac{\partial x_{i,f}}{\partial z}\right)|}_{z=h_{mth}}=0$
$i=H_{2}$	${\left(F_{f}\cdot x_{H_{2},f}-A_{mth}D_{H_{2},mth}\rho_{tot}\frac{\partial x_{H_{2},f}}{\partial z}\right)|}_{z=h_{mth}}=-\beta \cdot A_{mth}\rho_{tot}x_{H_{2},f}+\beta \cdot A_{m}\rho_{tot}x_{H_{2},dm}$
Gap with stagnate flow	
Retentate product flow $F_{r}$	${F_{r}=F_{f}|}_{z=h_{mth}}-F_{H_{2},perm}$ $,\;\mathrm{where}\;F_{H_{2},perm}=J_{dm,H_{2}}A_{m}=\beta \cdot \rho_{tot}A_{m}\left({\frac{A_{mth}}{A_{m}}x_{H_{2},f}|}_{z=h_{mth}}-x_{H_{2},dm}\right)$
Mole fractions in the retentate gas flow	
$i\neq H_{2}$	$x_{i,r}=\frac{{F_{f}x_{i,f}|}_{z=h_{mth}}}{{F_{f}|}_{z=h_{mth}}-F_{H_{2},perm}}$
$i=H_{2}$	$x_{H_{2},r}=\frac{{F_{f}x_{H_{2},f}|}_{z=h_{mth}}-F_{H_{2},perm}}{{F_{f}|}_{z=h_{mth}}-F_{H_{2},perm}}$

**Table 4 membranes-12-00741-t004:** Summary of the kinetic expressions used in the simulation.

Reactions			Rate Equations			
$C_{2}H_{5}\mathrm{OH}\;\to \;{\mathrm{CH}}_{4}\;+\;H_{2}\;+\;\mathrm{CO}\;\Delta_{r}H_{298\;K}^{o}\;=\;49.0\;\mathrm{kJ/mol}$			$r_{1}=k_{f,1}\cdot x_{\mathrm{Eth}}$			
${\mathrm{CH}}_{4}\;+\;H_{2}O\;↔\;3H_{2}\;+\;\mathrm{CO}\;\Delta_{r}H_{298\;K}^{o}=\;206.3\;\mathrm{kJ/mol}$			$r_{2}=k_{f,2}\cdot x_{{\mathrm{CH}}_{4}}^{\alpha_{2}}\cdot x_{H_{2}O}^{\beta_{2}}\cdot \left[1-\frac{Q_{2}}{K_{eq,2}}\right]$			
$\mathrm{CO}\;+\;H_{2}O\;↔\;{\mathrm{CO}}_{2}\;+\;H_{2}\;\Delta_{r}H_{298\;K}^{o}=\;-41.2\;\mathrm{kJ/mol}$			$r_{3}=k_{f,III}\cdot x_{\mathrm{CO}}^{\alpha_{3}}\cdot x_{H_{2}O}^{\beta_{3}}\cdot \left[1-\frac{Q_{3}}{K_{eq,3}}\right]$			
${\mathrm{CH}}_{4}\;+\;2H_{2}O\;↔\;4H_{2}\;+\;{\mathrm{CO}}_{2}\;\Delta_{r}H_{298\;K}^{o}=\;165.1\;\mathrm{kJ/mol}$			$r_{4}=k_{f,IV}x_{{\mathrm{CH}}_{4}}^{\alpha_{4}}\cdot x_{H_{2}O}^{\beta_{4}}\cdot \left[1-\frac{Q_{4}}{K_{eq,4}}\right]$			
Kinetic rate constant $k_{f,j}=A_{j}\cdot \exp\left(-\frac{E_{j}}{RT}\right)$			Equilibrium constant $K_{eq,j}=\exp\left(-\frac{\Delta G_{j}}{RT}\right)$		Reaction quotient for a reaction $c\;C+d\;D\;⇌\;a\;A+b\;B\;Q_{j}=\frac{{\left[p_{A}\right]}^{a}\cdot {\left[p_{B}\right]}^{b}}{{\left[p_{C}\right]}^{c}\cdot {\left[p_{D}\right]}^{d}}$	
Parameters of the rate equations for the reactions						
	$A_{j}\left(\frac{\mathrm{mol}}{m^{2}\cdot s}\right)$	$E_{j}\left(\frac{\mathrm{kJ}}{\mathrm{mol}}\right)$	$\alpha_{j}$	$\beta_{j}$	$K_{eq,j}$	Unit
$r_{1}$	1.4 × 10^4^	51	-	-		
$r_{2}$	1.86 × 10^5^	72	1	2	$1.167\times 10^{13}\exp\left(-\frac{26830}{T}\right)$	$\left({\mathrm{atm}}^{2}\right)$
$r_{3}$	4.08 × 10^4^	52	1	1	$1.767\times 10^{-2}\exp\left(\frac{4400}{T}\right)$	$\left({\mathrm{atm}}^{0}\right)$
$r_{4}$	1.408 × 10^4^	81	1	1.25	$2.062\times 10^{11}\exp\left(-\frac{22430}{T}\right)$	$\left({\mathrm{atm}}^{2}\right)$

**Table 5 membranes-12-00741-t005:** Constitutive equations for the feed-side reactor model.

Effective axial dispersion of mass in the monolith			
$D_{i,mth}=D_{i-mix}+\frac{u_{mth}{}^{2}d_{h,mth}{}^{2}}{192D_{i-mix}}$ $,\;\mathrm{where}\;D_{i-mix}=\frac{1-x_{i,f}}{\sum_{j=1,j\neq i}^{7}\frac{x_{j,f}}{D_{ij}}},$ $D_{ij}=\frac{10^{-7}T^{1.75}{\left[\left(\frac{1}{M_{i}}+\frac{1}{M_{j}}\right)\right]}^{\frac{1}{2}}}{P{\left[{\left(\sum \upsilon^{\prime }\right)}_{i}^{\frac{1}{3}}+{\left(\sum \upsilon^{\prime }\right)}_{j}^{\frac{1}{3}}\right]}^{2}}$			
Hydrogen mass transfer coefficient at feed-side of the membrane			
$\beta =\frac{Sh\cdot D_{i-mix}}{d_{h,m}}$ $,\;\mathrm{where}\;Sh=0.763Re^{0.5}Sc^{0.4}$ $;\;\delta_{h,m}=2.4\sqrt{\frac{\nu \left[\frac{m^{2}}{s}\right]}{u\left[\frac{1}{s}\right]}}=2.4\sqrt{\frac{\mu_{f}\cdot V_{gap}\cdot \rho_{tot}}{\rho_{f}\;\cdot F_{f}}}$			
$Re=\frac{\rho_{f}\cdot u\cdot d_{m}{}^{2}}{\mu_{f}}$ $\mathrm{and}\;Sc=\frac{\mu_{f}}{\rho_{f}\cdot D_{H_{2}-mix}},$			
$\mu_{f}=\sum_{i=1}^{n=7}\frac{x_{i}\mu_{i}}{\sum_{i=1}^{n=7}x_{j}\Phi_{ij}},$ $\mu_{i}=\mu_{i0}\frac{T_{0}+C_{i}}{T+C_{i}}{\left(\frac{T}{T_{0}}\right)}^{\frac{3}{2}},$		$\begin{array}{l} \Phi_{ij}=\frac{{\left[1+{\left(\frac{\mu_{i}}{\mu_{j}}\right)}^{\frac{1}{2}}{\left(\frac{M_{j}}{M_{i}}\right)}^{\frac{1}{4}}\right]}^{2}}{{\left[8\left(1+\frac{M_{i}}{M_{j}}\right)\right]}^{\frac{1}{2}}}, \end{array}$ $\Phi_{ji}=\Phi_{ij}\frac{\mu_{j}}{\mu_{i}}\cdot \frac{M_{i}}{M_{j}}.$	
The reference viscosity at reference temperature and Sutherland’s temperature for gaseous substances			
Substance	*C*, Sutherland’s temperature (K)	*T*_0_, reference temperature (K)	$\mu_{i0}$ reference viscosity (kg s^−1^ m^−1^)
H_2_O	673	873.16	3.09 × 10^−5^
CH_4_	164	873.16	2.46 × 10^−5^
CO_2_	240	873.16	3.61 × 10^−5^
CO	102	873.16	3.63 × 10^−5^
H_2_	72	873.16	1.83 × 10^−5^
Ar	142	873.16	4.87 × 10^−5^

**Table 6 membranes-12-00741-t006:** Governing and constitutive equations for the sweep-side reactor.

Dense layer of a thickness *h_dm_* in the membrane module		
Hydrogen flow at steady state $F_{H_{2},perm}=A_{m}Q_{dm}\left(\sqrt{P_{H_{2},dm}}-\sqrt{P_{H_{2},pl}}\right)$$,\;\mathrm{where}\;P_{H_{2},dm}=P\cdot x_{dm}$$\;\mathrm{and}\;P_{H_{2},pl}=P\cdot x_{pl}$ $\mathrm{Parity}\;\mathrm{flux}\;\mathrm{equation}\;\mathrm{at}\;\mathrm{the}\;\mathrm{feed}-\mathrm{side}\;\mathrm{boundary}\;A_{mth}\beta \cdot \rho_{tot}x_{H_{2},f}-A_{m}\beta \cdot \rho_{tot}x_{H_{2},dm}=A_{m}Q_{dm}\sqrt{P}\left(\sqrt{x_{H_{2},dm}}-\sqrt{x_{H_{2},pl}}\right).$		
Real solution of a quadratic from the parity equation for $x_{dm}$ $\sqrt{x_{H_{2},dm}}=\frac{-Q_{dm}\sqrt{P}+\sqrt{PQ_{dm}{}^{2}+4\beta^{2}\rho_{tot}{}^{2}\frac{A_{mth}}{A_{m}}{x_{H_{2},f}|}_{z=h_{mth}}+4\beta \rho_{tot}Q_{dm}\sqrt{P}\sqrt{x_{H_{2},pl}}}}{2\beta \rho_{tot}}$		
$\mathrm{Permeance}\;(\mathrm{mol}\;m^{-2}s^{-1}{\mathrm{atm}}^{-0.5})\;Q_{dm}=\frac{\Theta }{h_{dm}}\exp\left(-\frac{E_{dm}}{RT}\right).$	$\mathrm{Permeability}\;(\mathrm{mol}\;m^{-1}s^{-1}{\mathrm{atm}}^{-0.5})\;\Theta =1.4801\times 10^{-2}$	Activation energy (J mol^−1^) 60,000
Powder layer of the membrane module		
$\varepsilon_{pl}\rho_{tot}\frac{\partial x_{H_{2},pl}}{\partial t}-\rho_{tot}D_{H_{2}-\mathrm{Ar},pl}^{eff}\frac{\partial^{2}x_{H_{2},pl}}{\partial z^{2}}=0$ $\;\mathrm{and}\;x_{\mathrm{Ar},pl}=1-x_{H_{2},pl}.$		
Boundary conditions: at *z* = *h_mth_ + h*_gap_ *+ h_dm_*:	${-\rho_{tot}D_{H_{2}-Ar,pl}^{eff}\frac{\partial x_{H_{2},pl}}{\partial z}|}_{z=h_{mth}+h_{gap}+h_{dm}}=Q_{dm}\sqrt{P}\left(\sqrt{x_{H_{2},dm}}-\sqrt{x_{H_{2},pl}}\right),$	
at *z* = *h_mth_ + h*_gap_ *+ h_dm_ + h_pl_:*	$-\rho_{tot}D_{H_{2}-\mathrm{Ar},pl}^{eff}\frac{\partial x_{i,pl}}{\partial z}=\rho_{tot}D_{H_{2}-\mathrm{Ar},il}^{eff}\frac{\partial x_{i,il}}{\partial z}.$	
Intermediate layer of the membrane module		
$\varepsilon_{il}\rho_{tot}\frac{\partial x_{H_{2},il}}{\partial t}=\rho_{tot}D_{H_{2}-\mathrm{Ar},il}^{eff}\frac{\partial^{2}x_{H_{2},il}}{\partial z^{2}}$ $\;\mathrm{and}\;x_{H_{2},il}=1-x_{\mathrm{Ar},il}.$		
Boundary conditions: at *z* = *h_mth_ + h*_gap_ *+ h_dm_ + h_pl_*:	$-\rho_{tot}D_{H_{2}-\mathrm{Ar},il}^{eff}\frac{\partial x_{H_{2},il}}{\partial z}=\rho_{tot}D_{H_{2}-\mathrm{Ar},pl}^{eff}\frac{\partial x_{H_{2},pl}}{\partial z},$	
at *z* = *h_mth_ + h*_gap_ *+ h_dm_ + h_pl_ + h_il_:*	$-\rho_{tot}D_{H_{2}-\mathrm{Ar},il}^{eff}\frac{\partial x_{H_{2},il}}{\partial z}=\rho_{tot}D_{H_{2}-\mathrm{Ar},foam}^{eff}\frac{\partial x_{H_{2},foam}}{\partial z}.$	
$D_{H_{2}-\mathrm{Ar},pl(il)}^{eff}=D_{\mathrm{Ar}-H_{2},pl(il}^{eff}=\frac{\varepsilon_{pl(il)}}{\tau_{pl(il)}}\cdot \frac{1}{2}\left(\frac{1}{1/D_{H_{2},pl(il)}^{kn}+1/D_{H_{2}-\mathrm{Ar}}}+\frac{1}{1/D_{\mathrm{Ar},pl(il)}^{kn}+1/D_{H_{2}-\mathrm{Ar}}}\right).$		
$D_{i,pl(il)}^{kn}=\frac{d_{pore,pl(il)}}{3}{\left(\frac{8RT}{\pi M_{i}}\right)}^{1/2}=48.5d_{pore,pl(il)}\sqrt{\frac{T}{M_{i}}}.$	$d_{pore,pl(il)}=\frac{4\varepsilon_{pl(il)}}{S_{V,pl(il)}}.$	$d_{pore,pl(il)}=\frac{4\varepsilon_{pl(il)}}{S_{V,pl(il)}}.$
Foam layer in the membrane module		
$\varepsilon_{foam}\rho_{tot}\frac{\partial x_{H_{2},foam}}{\partial t}=\rho_{tot}D_{H_{2}-\mathrm{Ar},foam}^{eff}\frac{\partial^{2}x_{H_{2},foam}}{\partial z^{2}},$ $\;\mathrm{and}\;x_{\mathrm{Ar},foam}=1-x_{H_{2},foam},$		
$D_{H_{2}-\mathrm{Ar},foam}^{eff}=\varepsilon_{foam}\left(\frac{D_{H_{2}-\mathrm{Ar}}}{\tau_{foam}}+0.5d_{p,foam}u_{foam}\right),\;$where pore diameter $d_{p,foam}=\frac{d_{cell}}{\left(3.7033-2.5516\varepsilon_{foam}+0.7054\varepsilon_{foam}^{2}\right)},$		Tortuosity $\tau_{foam}=1+\frac{4.867\left[1-0.971{\left(1-\varepsilon_{foam}\right)}^{0.5}\right]}{4\varepsilon_{foam}{\left(1-\varepsilon_{foam}\right)}^{0.5}}\left(1-\varepsilon_{foam}\right).$
Boundary conditions: at *z* = *h_mth_ + h*_gap_ *+ h_dm_ + h_pl_ + h_il_*: at *z* = *h_mth_ + h*_gap_ *+ h_dm_ + h_pl_ + h_il_ + h_foam_:*	$-\rho_{tot}D_{H_{2}-\mathrm{Ar},foam}^{eff}\frac{\partial x_{H_{2},foam}}{\partial z}=\rho_{tot}D_{H_{2}-\mathrm{Ar},foam}^{eff}\frac{\partial x_{H_{2},il}}{\partial z},$ $-\rho_{tot}D_{H_{2}-\mathrm{Ar},foam}^{eff}\frac{\partial x_{H_{2},foam}}{\partial z}=\beta_{foam}\rho_{tot}\left(x_{H_{2}foam}-x_{H_{2},sw}\right).$	
Hydrogen effective mass transfer coefficient at the sweep-side		
$Sh_{d_{s,avg}}=\frac{\beta_{foam}\cdot d_{s,avg}}{D_{H_{2}-\mathrm{Ar},foam}^{eff}},$ $\;\mathrm{where}\;d_{s,avg}≅\frac{2.85\cdot \left(1-\varepsilon_{foam}\right)}{S_{V,foam}},$ $\;\mathrm{and}\;Sh_{d_{s,avg}}=\varepsilon_{foam}^{-2}\left(0.566Re_{d_{s,avg}}^{0.33}+0.039Re_{d_{s,avg}}^{0.8}\right)Sc_{foam}^{1/3}$		
$Re_{s,avg}=\frac{\rho_{foam}\cdot u_{foam}\cdot d_{s,avg}}{\mu_{foam}},$	$Sc_{foam}=\frac{\mu_{foam}}{\rho_{foam}\cdot D_{H_{2}-\mathrm{Ar},foam}^{eff}},$	$S_{V,foam}=\left(\frac{2\sqrt{3\pi }}{d_{cell}}\right){\left(1-\varepsilon_{foam}\right)}^{0.5}.$
Sweep compartment		
Volumetric gas flow rate $G_{sw}=G_{sw}^{0}+\beta_{foam}A_{m}\left({x_{H_{2},foam}|}_{z=h_{mth}+h_{gap}+h_{dm}+h_{pl}+h_{il}+h_{foam}}-x_{H_{2},sw}\right)$ and $x_{H_{2},sw}+x_{\mathrm{Ar},\mathrm{sw}}=1$		
$x_{H_{2},sw}=\frac{\left(\beta_{foam}+\frac{G_{sw}^{0}}{A_{m}}+\beta_{foam}\right)-\sqrt{{\left(\beta_{foam}+\frac{G_{sw}^{0}}{A_{m}}+\beta_{foam}\cdot x_{H_{2},foam}\right)}^{2}-4\beta_{foam}x_{H_{2},foam}}}{2\beta_{foam}}.$		

## Data Availability

Not applicable.

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
