# Peer review of "Model-Based Performance Analysis of Membrane Reactor with Ethanol Steam Reforming over a Monolith"

_membranes, 2022, doi:10.3390/membranes12080741_

Round 1

Reviewer 1 Report

The manuscript deals with the ethanol steam reforming reaction carried out in membrane reactors to generate high grade hydrogen. The topic is not novel, but the work is intersting, even though it needs severe revisions. In particular, several part of the text should be revised in order not to provide misleading concepts.

- Firstly, the utilization of Catalytic Membrane Reactors is not appropriate in this case, because a membrane reactor is defined "catalytic" when the membrane itself plays the role of a catalyst during such a reaction (that means that no catalyst is packed in the membrane reactor). In my opinion, in this work the Authors used simply a Membrane Reactor.

- In Introduction section, the following statement is not true: "The driving force for hydrogen transmembrane permeation at the ambient pressure is a hydrogen partial pressure gradient at the sides of the membrane." The hydrogen permeation driving force is defined as hydrogen partial pressure difference across the membrane, at any pressure! Not only at ambient pressure...this detail is fully misleading.

- The state of the art regarding the membrane engineering in process intensification, membrane reactors and ethanol steam reforming in membrane reactors is missing or not adequate. In particular, ref. [3] is too old. Place replace with "Drioli et al., Process intensification strategies and membrane engineering, Green Chem. 14 (2012) 1561-1572" or some other more appropriate references; References [4,7,9,10,11,13] are not adequate because describing process not for hydrogen production/separation or because the topic is a bit far from the hydrogen perm-selective membranes. I suggest to insert an adequate literature on ethanol steam reforming reaction in membrane reactors as reported below:

1) Amiri et al., Membrane reactors for sustainable hydrogen production through steam reforming of hydrocarbons, Chemical Engineering and Processing: Process Intensification, 157 (2020) 108148.

2) Iulianelli et al., From bioethanol exploitation to high grade hydrogen generation: steam reforming promoted by a Co-Pt catalyst in a Pd-based membrane reactor, Renewable Energy, 119 (2018) 834-843.

3) Iulianelli, S. Liguori, A. Vita, C. Italiano, C. Fabiano, Y. Huang, A. Basile, The oncoming energy vector: hydrogen produced in Pd-composite membrane reactor via bioethanol reforming over Ni/CeO2 catalyst, Catalysis Today, 259 (2016) 368-375.

4) Basile et al., Ethanol steam reforming reaction in a porous stainless steel supported palladium membrane reactor, International Journal of Hydrogen Energy, 36 (2011) 2029 -2037.

5) Iulianelli et al., Hydrogen production from ethanol via inorganic membrane reactors technology: a review, Catalysis Science & Technology, 1 (2011) 366-379.

6) Alique et al., Ultra-Pure Hydrogen via Co-Valorization of Olive Mill Wastewater and Bioethanol in Pd-Membrane Reactors, Processes, 8 (2020) 219.

7) Chen et al., Reaction and hydrogen production phenomena of ethanol steam reforming in a catalytic membrane reactor, Energy, 220 (2021) 119737.

8) Mironova et al., Hydrogen Production by Ethanol Steam Reforming in the Presence of Pd-, Pt-, Ru-, and Ni-Containing Nanodiamonds in Conventional and Membrane Reactors, Membranes and Membrane Technologies, pages(2019) 246–253 .

- 8 self-citations over total 54 references is too much. 

- In Introduction section, the Authors should clearly point out the importance of the catalyst choice, and what kind of ethanol reforming effective catalysts have been used in literature as well.

- The description of the state of the art should not be limited to the previous papers published by the Authors, but it should highlight what is the novelty of the present one with respect to a wider literature.

-  In Table 1, it is reported that the membrane thickness is equal to 150 micro-meters. Does it refer to the supported membrane described in Par. 2.1.2? If this is the case, the Authors should well point out the dimension of the separative layer, the intermediate layer (if any) and the substrate as well.

- In Par. 2..2, th Authors assume that the membrane follows Sieverts-Fick law; then, they assume that it is fully hydrogen perm-selective. Did they check it under real experimental tests?

- I am not sure that this Reviewer well understood the assumption nr. 6 in Par. 2.2: "Molecular and Кnudsen diffusions control the transport through the 241 gas-permeable layers of the membrane module." Why? If at the point nr. 5, they assume that Sieverts-Fick law is controlling the hydrogen permeation through the membrane, why did they assume also the control of Knudsen and molecular transport mechanism? The latter assumption could be realistic if the membrane contains defects and the other gases a part from hydrogen pass across the membrane by such mechanisms. If this is the meaning of this assumption, consequently assumption nr. 5 is unrealistic because the hydrogen partial pressure exponent (n) should be not equal to 0.5 (Sieverts), but it should be assessed by the graph H2 permeating flux vs transmembrane hydrogen partial pressure (at different n values), evaluating the maximum linear regression factor (R2) at each n value established. For more details, please, follow the indications reported in: Iulianelli et al., A supported Pd-Cu/Al2O3 membrane from solvated metal atoms for hydrogen separation/purification, Fuel Processing Technology, 195 (2019) 106141-106149

Is it realistic to operate a membrane reactor at 900 °C? Is it convenient from an energy saving point of view? Many other membrane reactor applications run at much lower temperature. What is the advantage of such a technical solution?

- The Authors should well point out the reference of the values used in Table 5 regarding hydrogen permeance, permeability and activation energy.

- Fig. 4b: more experimental data should be reported in the graph.

- The error bar should be reported in all the experimental points plotted in the various graphs of this work.

- The Authors should clearly point out the purity of the hydrogen collected in the permeate side, further adding a graph in which it is evidenced the hydrogen recovered for permeation through the membrane over the total produced.

- What about coke deposition?

Author Response

Please see  the attachment "Reply_to_Reviewer1"

Reviewer 2 Report

This paper uses 1D isothermal reaction transport model to analyze performance of catalytic membrane reactor with ethanol. I opined that the main finding of the abstract and conclusion to be rewritten concisely and in a more confident manner. What are the "exact" new knowledge? This can be drawn given that there are a lot of discussion and data found from the manuscript. Another concern is that this model is only applied to lab scale? How about the large-scale modelling? What is the limitation of the current work?

Author Response

Reply to  Reviewer2

This paper uses 1D isothermal reaction transport model to analyze performance of catalytic membrane reactor with ethanol. I opined that the main finding of the abstract and conclusion to be rewritten concisely and in a more confident manner. What are the "exact" new knowledge? This can be drawn given that there are a lot of discussion and data found from the manuscript. Another concern is that this model is only applied to lab scale? How about the large-scale modelling? What is the limitation of the current work?

Reply:

The abstract and conclusion have been rewritten.

Scale up of hydrogen membrane technologies is the most challenging yet important task for all researchers. The following can be said about scaling. The reliability and validity of the modeling tool, and whether we can get more out of our model by tuning the model parameters, is the most important issue when up-scaling. The quality and significance of experiments play a crucial role in this question. The applicability of the one-dimensional model formulation and computational simulations are verified and validated by direct comparison of model results with existing measurements of interest. The aim was to explain the better performance in the case of a monolith-assisted MR with the same membrane module used in a packed MR. Of course, using only one set of validation data has some drawbacks being not enough to perform large scale simulations. Design parameters also play an important role, and they must be taken into account in each specific case. More experimental studies of membrane reactor as well as computational work would be vital to the design of such membrane reactor systems for an industrial application.

Reviewer 3 Report

The manuscript under revision addresses a one-dimensional mathematical model to describe the behavior of a membrane reactor to produce hydrogen through ethanol steam reforming and containing the catalyst in a monolite structure. The topic is certainly interesting and the manuscript is, in general, well-written in terms of english style and scientific sound. However, some important points should be clarified, expanded or corrected prior to take a formal decision about its potential publication. Next, I include the most important items to be discussed:

1. The abstract section should include more relevant results extracted from the research, as well as the main insights of the work.

2. Introduction section. The state of the art is not deep enough, basically describing previous results reached by a particular research group. More information about the H2 production process, catalyst systems and membranes need to be Included. In this context, only a 37% of references be,ong to the last 5 years. More recent works should be cited.

3. Introduction section. In paragraph written from lines 60-69, authors talk about PBMR but later indicate "the membrane was placed inmediately downstream from PB...". Then, it is not a real membrane reactor. Please, discuss and clarify.

4. Introduction section. Objetive of the work. Interest of the work related with the current state of the art is not clear. Please, highlight the interest of your work. Related to that, discuss the utility of a one-dimensional model, the most possible simple one, in contrast to other containing more dimensions.

5. More details about the considered membrane should be included, not only geometrical values.

6. Ranges for operating conditions in the reforming reaction and H2 separation should be further justified. Why these temperatures? why these pressures? why this water/ethanol ratio? why the use of sweep gas if the main tarjet is to obtain pure hydrogen?

7. Figure 3. More details about the membrane should be included, indentifying each layer on the images and expanding the very brief description given in the related text.

8. Model assumptions Points 3 and 4 should be further explained. Are they logical? approximate values?

9. Chemical reactions considered. Have you considered side or undesiderable reactions (i.e. coke formation)?

10. Validation of modeling results with experimental data. This section is confusing. More details should be given for each particular case. Source of experimental results used for validation needs to be discussed. The membrane permeability is indicated, but what is about the activation energy or other adjusting parameters?

11. H2 recovery of around 12% is very low. How could you modify the operating parameters of your reactor to maximize this value?

12. Figures 8 and 9 are not clear enough. Difficult to understand all the information. Please, reconsider the use of approppiatte legends and eexpand the related explanation.

13. Conclusions should be reformulated including more concrete quantitative results reached from the study.

Author Response

Reply to Reviewer3

Comments and Suggestions for Authors

The manuscript under revision addresses a one-dimensional mathematical model to describe the behavior of a membrane reactor to produce hydrogen through ethanol steam reforming and containing the catalyst in a monolite structure. The topic is certainly interesting and the manuscript is, in general, well-written in terms of english style and scientific sound. However, some important points should be clarified, expanded or corrected prior to take a formal decision about its potential publication. Next, I include the most important items to be discussed:

1. The abstract section should include more relevant results extracted from the research, as well as the main insights of the work.

Reply:

Abstract has been extended.

2. Introduction section. The state of the art is not deep enough, basically describing previous results reached by a particular research group. More information about the H2 production process, catalyst systems and membranes need to be Included. In this context, only a 37% of references be,ong to the last 5 years. More recent works should be cited.

Reply:

Corrected. Recent works (last 5 years) have been cited.

3. Introduction section. In paragraph written from lines 60-69, authors talk about PBMR but later indicate "the membrane was placed immediately downstream from PB...". Then, it is not a real membrane reactor. Please, discuss and clarify.

Reply:

The Introduction section has been rewritten

4. Introduction section. Objetive of the work. Interest of the work related with the current state of the art is not clear. Please, highlight the interest of your work. Related to that, discuss the utility of a one-dimensional model, the most possible simple one, in contrast to other containing more dimensions.

Reply:

The utility of a one-dimensional model applied for a MR with the disc-shaped membrane was discussed in a previous paper published by the authors [Bobrova, L.; Eremeev, N.; Vernikovskaya, N.; Sadykov, V.; Smorygo, O. Effect of asymmetric membrane structure on hydrogen transport resistance and performance of a catalytic membrane reactor for ethanol steam reforming. Membranes 2021, 11, 332.]. Thus, hydrodynamics of flow and effect of operational factors on fluid dynamics in a similar reactor design were studied by using the computational fluid dynamic simulations in the following papers [Hong, J.; Kirchen, P.; Ghoniem, A. Numerical simulation of ion transport membrane reactors: Oxygen permeation and transport  and fuel conversion. J. Membr. Sci. 2012, 407–408, 71–85.

Gozá½¹lvez-Zafrilla, J.M.; Santafé-Moros, A.; Escolástico, S.; Serra, J.M. Fluid dynamic modeling of oxygen permeation through mixed ionic–electronic conducting membranes. J. Membr. Sci. 2011, 378, 290–300.]. It was concluded that if the thickness of the membrane disk module is small compared with its diameter, deflection of flow streamlines and concentration gradients in radial direction, at the proximity of the membrane, is rather low, while the gradients in the axial direction are sufficient. Therefore, a significant hydrogen-partial-pressure gradient exists only across the membrane, and due to the radial symmetry, the flow can be considered as one-dimensional. With the purpose to capture concentration distributions in axial direction, a computationally efficient 1-D modeling approach has been applied, while the diffusion along radius could be neglected. The flow field in the feed compartment is coupled with the membrane module in its one-dimensional form.

The Introduction section has been rewritten. The objective of the study is specified  as following:

The research concept of this paper, based on the intensive experimental study, is to investigate, by conducting a model-based performance analysis, all the main phenomena that determine the performance of the monolithic catalyst –assisted MR, namely design parameters, chemistry, transfer effects, and hydrogen permeation across the membrane layers, as well as boundary layer effects. As far as we know, we have not found such a study in the available literature.

5. More details about the considered membrane should be included, not only geometrical values.

Reply:

Done

6. Ranges for operating conditions in the reforming reaction and H2 separation should be further justified. Why these temperatures? why these pressures? why this water/ethanol ratio? why the use of sweep gas if the main tarjet is to obtain pure hydrogen?

Reply:

The optimal conditions of the reaction (including temperature, partial pressures, water/ethanol ratio) were selected according to the experimental study with the  published results:

Bobrova, L.; Eremeev, N.; Vernikovskaya, N.; Sadykov, V.; Smorygo, O. Effect of asymmetric membrane structure on hydrogen transport resistance and performance of a catalytic membrane reactor for ethanol steam reforming. Membranes 2021, 11, 332.

Eremeev, N.; Krasnov, A.; Bespalko, Yu,; Bobrova, L.; Smorygo, O.; Sadykov, V. An experimental performance study of a catalytic membrane reactor for ethanol steam reforming over a metal honeycomb catalyst. Membranes 2021, 11, 790.

Sweep gas (argon) was used to decrease hydrogen partial pressure at the permeate side of the membrane and, hence, to increase driving force of hydrogen permeation.

7. Figure 3. More details about the membrane should be included, indentifying each layer on the images and expanding the very brief description given in the related text.

Reply:

Details of the membrane including description, images and table with each layer parameters have been added.

8. Model assumptions Points 3 and 4 should be further explained. Are they logical? approximate values?

Reply:

Model assumptions have been changed to the following:

Both the reaction and transport are conducted isothermally. This assumption for the endothermic reaction of ethanol reforming signifies that sufficient heat is supplied to maintain an approximately constant reaction temperature.

The internal mass transfer resistance of the catalytic washcoat over the metallic monolith is neglected.

Negligible pressure drop occurs across the membrane reactor.

The permeation of hydrogen through the dense layer of the membrane module follows Sievert's law.

Molecular and Кnudsen diffusions control in the hydrogen and argon transport through the gas-permeable layers of the membrane support.

A perfect-gas equation of state relates the density, pressure, temperature, and composition.

Approximate values of effective mass dispersion coefficients are obtained to be in the range of ≈ 6∙10-4 to 1∙10-2 –m2 s‑1 for Pe=0.76

9. Chemical reactions considered. Have you considered side or undesiderable reactions (i.e. coke formation)?

Reply:

In experimental studies, a noticeable effect of the ethanol concentration on the carbon yield correlated with a change in the space velocity, i.e. the ratio of the  flow rate of reactants to the catalyst volume. An increase in carbon imbalance under operating conditions with a high fuel concentration, at low temperature, high flow rates, high partial pressure of ethanol in the feed, is probably a consequence of the conversion of ethanol into reaction products - coke precursors.  Therefore, based on the influence of operating parameters such as temperature, fuel composition (steam to ethanol molar ratio) and reactant concentrations, both the reactor operation and regeneration procedures must be optimized to solve this problem.

However, it should be noted that under optimized conditions (as in the case of comparison), coke deposition was not observed both on the membrane surface and on the monolithic catalyst, which is consistent with the stable operation of the membrane reactor. These conditions are met for the reference case

10. Validation of modeling results with experimental data. This section is confusing. More details should be given for each particular case. Source of experimental results used for validation needs to be discussed. The membrane permeability is indicated, but what is about the activation energy or other adjusting parameters?

Reply:

Actually,  for this membrane module, the mathematical model and adjusting parameters were described in detail in a previous paper published by the authors [Bobrova, L.; Eremeev, N.; Vernikovskaya, N.; Sadykov, V.; Smorygo, O. Effect of asymmetric membrane structure on hydrogen transport resistance and performance of a catalytic membrane reactor for ethanol steam reforming. Membranes 2021, 11, 332.]. The one-dimensional isothermal reaction-transport model for the MR with a packed bed was confirmed by the performance data  from a lab-scale reactor with the same disk-shaped membrane.

As indicated in the text of the paper, the parameters in the expression for the hydrogen permeation flow rate, apparent activation energy (60 kJ/mol), and  the order of magnitude of permeability were obtained from study of  the temperature-dependent conductivity [Uvarov, N.F.; Ulichin, A.C.; Bespalko, Yu.N.; Eremeeev, N.F.; Krasnov, A.V.; Skriabin, P.I.; Sadykov, V.A. Study of proton conductivity of composite metal-ceramic materials based on neodimium tugstates using a four-electrode technique with ionic probes. Int. J. Hydrog. Energy 2018, 43(42), 19521–19527.]. The permeability value of hydrogen  is fitted to the experimental data of reference case (Table 1) and is    mol m-1s-1atm-0.5 (Table 5).

The reference [Uvarov, N.F.; Ulichin, A.C.; Bespalko, Yu.N.; Eremeeev, N.F.; Krasnov, A.V.; Skriabin, P.I.; Sadykov, V.A. Study of proton conductivity of composite metal-ceramic materials based on neodimium tugstates using a four-electrode technique with ionic probes. Int. J. Hydrog. Energy 2018, 43(42), 19521–19527.] was added.  Par. 2.2. Model Description, supplemented with data on the structure of the membrane

H2 recovery of around 12% is very low. How could you modify the operating parameters of your reactor to maximize this value?

Reply:

It was a mistake. The value of 12.12 % is simply the percentage molar ratio of hydrogen penetrated through the membrane to the total flow after the catalyst.

As for the estimation of the hydrogen recovery coefficient, which is calculated as the molar ratio of hydrogen passed through the membrane to hydrogen obtained as a result of the ethanol steam reforming  reaction, as well as the percentage yield as the degree of achievement of the theoretical yield of the reaction, they were analyzed in the previous article. Here, in the figure, we show the efficiency characteristics of the  membrane reactor under the study

Instead of the wrong sentence, the following is inserted:

The percentage values of hydrogen recovery and yield estimated by experiments at 800°C give 47% and 44%, respectively.

In addition, the first paragraph in Par. 3.1.2 has been rewritten:

Generally, comparative analysis of CMRs is performed by using characteristics of recovery, yield of hydrogen, fuel conversion and resistances to internal mass transfer of a permeating component. The details of introducing a structural catalyst in the CMR reactor instead of a packed bed on the overall efficiency of the integrated reaction-separation process were discussed in detail in the previous paper [21]. It was shown that the reactor operating with the catalytic monolith showed better performance in terms of both hydrogen recovery and the yield with respect to the packed bed catalyst. Thus, a comparative study showed that in the case of a packed bed, an increase in temperature from 700oC to 900oC leads to an increase in the yield of hydrogen by almost 40% (from 24% to 33%), while the yield for the monolith practically does not change and averaged 37%. Hydrogen recovery for the monolith was about 10% higher in all operating temperatures

12. Figures 8 and 9 are not clear enough. Difficult to understand all the information. Please, reconsider the use of approppiatte legends and expand the related explanation.

Reply:

-          In Fig.8 (a), the description of the axis has been changed.

-          The following changes in the text

Figure 8. Effect of the distance above the membrane surface (a) and the feed flow rate (b) on the permeation phenomena.

We modeled various distances between the monolithic catalyst and the membrane surface, as well as feed rates, to determine their effect on permeability characteristics. As expected, the thickness of the boundary layer affects hydrogen diffusion rate through. Figure 8a, left panel, shows that by enlarging the gap above the membrane surface from 4 to 15 mm, the thickness of the boundary layer increases from 1.28 mm to 2.48 mm at 700oC, and from 1.46 mm to 2.83 mm at 900oC. This leads to a decrease in the mass transfer velocity of hydrogen β (at 700oC from 0.35 to 0.09 m/s, at 900oC from 0.81 to 0.21 m/s) with a corresponding decrease in the hydrogen flux (about 3 -4%). Indeed, a smaller distance between the monolith and the membrane can lead to compressing the boundary layer and thereby provide better diffusion of hydrogen into the dense layer.

Higher feed flow rates have the same effect on the boundary layer: the thickness of the boundary layer on feed-side decreases. However, this leads to lower values of both the hydrogen flux and hydrogen concentration in the sweep compartment, as shown in Figure 8b, due to an increase in hydrogen concentration in the retentate flow coming out of the gap space. And the higher the feed rate, the higher the negative effect of a larger gap height above the membrane surface can be.

-          In the case of Figure 9, the explanation has been changed to the following:

 The higher hydrogen flux in the case of the MR with a monolith may also be due to  the influence of the boundary layer phenomenon after the impingent of the feed stream (product gas from catalytic monolith), which affects the corresponding driving forces of   interfacial hydrogen transfer rate in the membrane. The effect of the operating temperatures on the driving force, expressed here as the difference between concentrations on each side of a boundary layer adjacent to the membrane surface and magnitude of the corresponding permeation fluxes is shown in Figure 9. It can be seen that in the case of stagnate boundary layer, the driving force is about an order of magnitude higher, and accordingly the permeation flux is 30-40 % greater. It is obvious that the hydrogen flux through the membrane is dictated by the number of molecules impinging and interacting with the interfacial surface under the boundary condition being formed.

 At the same time, the stagnate boundary layer phenomenon must be properly taken into account for the specific reactor configuration and operating conditions, otherwise incorrect conclusions about the technological parameters will be drawn. For example, if the feed flow is moving at a very high speed, it may have some kinetic energy, and this may lead to undesirable effects on the penetration rate.

13. Conclusions should be reformulated including more concrete quantitative results reached from the study.

Reply:

Conclusions have been  reformulated including more concrete quantitative results reached from the study.

Round 2

Reviewer 1 Report

The manuscript has been sufficiently improved by the Authors, as requested. Now, it can be accepted for publication.